# A Hybrid Type II Hub-and-Spoke Model Evaluation Framework in the Commonwealth Partnerships for Antimicrobial Stewardship Programme—A Study Protocol

**DOI:** 10.3390/antibiotics14121218

**Published:** 2025-12-03

**Authors:** Ayesha Iqbal, Gizem Gülpinar, Claire Brandish, Maxencia Nabiryo, Frances Garraghan, Victoria Rutter

**Affiliations:** 1Commonwealth Pharmacists Association, London E1W 1AW, UK or gizemgulpinar@gazi.edu.tr (G.G.); claire.brandish@commonwealthpharmacy.org (C.B.); maxencia.nabiryo@commonwealthpharmacy.org (M.N.); fgarraghan@bsac.org.uk (F.G.); victoria.rutter@commonwealthpharmacy.org (V.R.); 2Office of Lifelong Learning and the Physician Learning Program, Faculty of Medicine and Dentistry, University of Alberta, Edmonton, AB T6G1C9, Canada; 3Department of Pharmacy Management, Faculty of Pharmacy, Gazi University, Ankara 06330, Türkiye; 4Buckinghamshire Healthcare NHS Trust, Stoke Mandeville Hospital, Mandeville Road, Aylesbury HP21 8AL, UK; 5Microbiology Department, Manchester University NHS Foundation Trust, Cobbett House Manchester Royal Infirmary, Oxford Road, Manchester M13 9WL, UK

**Keywords:** antimicrobial stewardship, commonwealth partnerships for antimicrobial stewardship, hub-and-spoke model, healthcare, implementation science, evaluation framework

## Abstract

**Introduction:** The hub-and-spoke model (HSM) offers a methodological and hierarchical project management framework for efficient healthcare service delivery. The Commonwealth Partnerships for Antimicrobial Stewardship (CwPAMS) programme supports the development and implementation of antimicrobial stewardship (AMS) interventions to optimise the use of antimicrobials across eight African countries: Ghana, Kenya, Malawi, Nigeria, Sierra Leone, Tanzania, Uganda, and Zambia. Currently the second phase of the programme (CwPAMS 2.0) is being implemented, between March 2023 and March 2025, in which six countries are adopting the HSM to deliver AMS interventions. The aim of this study was to design a hybrid II monitoring, evaluation and learning (MEL) framework to assess the implementation and effectiveness of the HSM in delivering and adapting AMS interventions. **Methods:** A mixed-methods Hybrid II Implementation trial design was used to develop the MEL framework, guided by the Theory of Change, Socio-Ecological Theory, and Normalisation Process Theory and in alignment with the Reach, Effectiveness, Adoption, Implementation, and Maintenance (RE-AIM) framework. Data collection will be continuous and longitudinal (pre-, mid- and post-implementation). The evaluation framework has been designed to study implementation evaluation at three ecological levels: programme, partnership, and individual site level. Data collection will encompass mixed methodologies and include non-participant observations, formal and informal feedback (from individual key stakeholders and groups), knowledge tools and surveys, scored evaluations, pre-and post-assessments, semi-structured interviews, focus groups, and data collected during formal and informal meetings. This approach will facilitate continuous data collection for evaluation and help study the utilisation and adaptation of AMS interventions. **Discussion:** This study provides a protocol for developing and utilising an MEL framework to study the application of the HSM in delivering AMS interventions. Developing an evaluation framework requires meticulous planning and a robust implementation and evaluation protocol to ensure methodological rigour, transparency, and effective resource management throughout the project lifecycle. Despite comprehensive consideration of developing progress and programmatic indicators and measures across all domains, the study acknowledges limitations in definitively attributing causality to individual AMS interventions due to their complexity and varied implementation contexts.

## 1. Introduction

Clinical and health services research has consistently highlighted a persistent issue: the failure to translate research findings into practical applications and policy changes [1]. Even with significant annual investments amounting to billions of dollars from public and private sectors in biomedical research, clinical training, quality enhancement, and patient safety, healthcare systems and interventions continue to face challenges in surmounting the ‘quality chasm’ [2,3]. Demonstrating an innovation’s effectiveness is insufficient for adoption—implementation must align with both the ecosystem’s context and adopter characteristics to achieve sustained usage [4,5,6].

For interventions to succeed, effective implementation is crucial. Implementation research examines how interventions are put into practice and considers the factors that help or hinder their effectiveness. By understanding these processes and contexts, it provides insights to improve the adoption of evidence-based healthcare interventions [7]. It encompasses diverse theories, models, and frameworks aimed at enhancing the applicability of findings across different settings, contributing to the sustainability and scalability of interventions [8]. Frameworks within the domain of implementation science offer systematic approaches to evaluation. They also help address challenges by fostering consensus among stakeholders on programme understanding, evaluation procedures, and suitable goals and methodologies. Implementation frameworks such as RE-AIM (Reach, Effectiveness, Adoption, Implementation, Maintenance) can play a crucial role in understanding and assessing the implementation outcomes of an intervention [9]. However, it is important to note that RE-AIM does not elucidate the factors that shape implementation success; it can only confirm whether the implementation was carried out correctly [9,10]. Thus, to understand context, an implementation science determinant framework can help inform the enablers and barriers impacting implementation [8]. Consolidated Framework for Implementation Research (CFIR) is a conceptual framework developed to guide the systematic assessment of multi-socioecological level implementation contexts to identify factors that might influence interventional or implementational effectiveness. CFIR provides a framework of constructs arranged across five domains. It is a practical, theory-based guide for systematically assessing potential contextual determinants, which not only enhances understanding of the contextual factors affecting implementation outcomes, but also informs the development of targeted strategies to optimise the adoption and sustainability of evidence-based practices and interventions in diverse healthcare settings [11,12].

Antimicrobial resistance (AMR) poses a global public health challenge that can be mitigated through enhanced knowledge and implementation of antimicrobial stewardship (AMS) practices [13]. The Commonwealth Partnerships for Antimicrobial Stewardship (CwPAMS) is an initiative supported by the UK Department of Health and Social Care’s (DHSC) Fleming Fund and administered by the Commonwealth Pharmacists’ Association (CPA) and Global Health Partnerships (GHP). Launched in 2019 with 12 health partnerships (HPs) between UK and African institutions, the primary objective of the CwPAMS 1.0 programme was to strengthen the capacity of healthcare institutions and professionals in four low-and middle-income countries (LMICs) in the Commonwealth (CW) to tackle the challenges of AMR [14]. In 2021, it expanded (CwPAMS 1.5) to assist 14 HPs in bolstering AMS capabilities across eight African CW countries. In March 2023, CwPAMS 2.0 was introduced, comprising 24 HPs spanning eight CW countries: Ghana, Kenya, Malawi, Nigeria, Sierra Leone, Tanzania, Uganda and Zambia. The programme aims to support interventions that enhance the AMS capabilities of health institutions, reducing the risk of challenges posed by AMR.

Phase two of the programme (CwPAMS 2.0) implemented a hub-and-spoke model (HSM) to optimise AMS interventions across healthcare facilities. This strategic approach was adopted in six HPs, effectively extending their reach to 34 affiliated spoke facilities. This model capitalises on resources, improves access to care, and ensures consistency in service quality across the network [15]. The HSM has been successfully used for hepatitis, dispensing in community pharmacies, sexual health services, and more recently for gene therapies and rarer conditions [15,16]. Figure 1 illustrates the HSM setup in this study.

This study presents our hybrid evaluation framework, designed to assess the implementation and effectiveness of the HSM in delivering AMS interventions across HPs for the CwPAMS 2.0 programme. This encompasses two domains: first, an ongoing process evaluation of HSM feasibility within the CwPAMS programme, utilising the RE-AIM framework to assess implementation outcomes and identify barriers and facilitators at pre-, mid-, and post-implementation stages; second, an ongoing evaluation of HSM effectiveness in achieving CwPAMS interventional outcomes, measuring progress towards programmatic goals through the development of both mandatory and non-mandatory key performance indicators.

## 2. Methodology

### 2.1. Study Design and Conceptual Framework

A mixed method implementation hybrid II research design [17] will be adopted in this study as shown in Figure 2.

A mixed methods approach has been selected since the monitoring and evaluation of the CwPAMS programme cannot be conducted using one method. In addition to the RE-AIM framework and CFIR, the conceptual underpinnings of this study arise from the theory of change [18], socio-ecological theory [19] and normalisation process theory [20]. The conceptual framework has guided the development of two log-frames in this study: programmatic progress indicators and implementation progress indicators.

### 2.2. Data Collection

Data collection for this study will span March 2023 to March 2025, employing multiple assessment methods to track progress against log-frame indicators. While primarily designed for virtual administration to participants, the data collection tools will also accommodate in-person observations and event-based gatherings when feasible. Data collection began at the programme inception (March 2023) and continued prospectively through three distinct phases: pre-implementation (baseline, March–June 2023), mid-implementation (ongoing assessment, July 2023–December 2024), and post-implementation (March 2025). This prospective approach enables real-time monitoring, adaptive management of implementation challenges, and longitudinal tracking of changes. The study design is consistent with hybrid type II implementation trials, which emphasize dual testing of implementation strategies alongside intervention effectiveness. The following tools will be developed:

#### 2.2.1. Monitoring and Evaluation (MEL) Web Portal

A monitoring, evaluation and learning web portal (MEL-portal) will be used in CwPAMS 2.0 to obtain periodic updates on each HP’s progress. The MEL portal will collect quantitative information on programmatic log-frame indicators. Each HP will be granted access to the MEL portal and will provide data specific to evaluation indicators as an update on their progress or milestones (outputs, outcomes) achieved, if any.

#### 2.2.2. Surveys

Surveys will be designed by individual partnerships and used to capture and report progress against mostly quantitative data and some qualitative data through open ended questions. Examples include pre- and post-assessments specific to interventions, Knowledge, Attitude and Perception (KAP) surveys, intervention feedback evaluation surveys, monitoring and evaluation surveys, implementation evaluation surveys, AMS workstream specific surveys (leadership and training, knowledge and awareness of substandard and falsified medicines, microbiology, etc.) and behaviour change surveys (Appendix A).

#### 2.2.3. Data Collection Forms

In addition to surveys, specific data collection forms have been designed to capture uniform qualitative data across all HPs; examples include an AMS pre- and post-assessment tool, quarterly narrative report template and implementation evaluation survey (exemplar template tools have been provided in the Appendix A).

#### 2.2.4. Interviews and/or Focus Groups

In-depth qualitative data will be collected through interviews and/or focus groups with key stakeholders using semi-structured topic guides designed specifically for this project. These will be undertaken virtually or in-person depending on feasibility and the mutual convenience of the participants and the interviewing team. An example interview guide is provided in the Appendix A.

#### 2.2.5. Observational Visits

Periodic visits by CwPAMS technical and programmatic teams (including grant managers, technical experts and researchers) to each participating country will take place. These visits will likely involve meetings with HPs to gather real-time observational data on intervention and implementation progress based on predefined indicators. Qualitative data will be systematically collected by in-person visit reports. Additional data will be collected to identify exceptional cases or instances of exemplary practice, which will serve as valuable case studies for further analysis and learning.

#### 2.2.6. Exclusion Criteria

All data collected will be screened against predefined inclusion criteria aligned with our log-frame indicators (Table 1) and RE-AIM framework domains. Data will be considered irrelevant and excluded if (1) the data do not align with any of the mandatory or non-mandatory outcome measures defined in our evaluation framework, or (2) lack sufficient detail to contribute meaningfully to our implementation evaluation. The research team (AI and GG) will review all collected data during monthly meetings to ensure alignment with study objectives, and any ambiguous cases or unusual data will be discussed with the broader team for consensus decision-making.

### 2.3. Sampling Framework

Surveys and data collection tools will be sent to relevant stakeholders. Periodic reminders will be sent to improve response rate and data collection. For interviews and focus groups, purposeful maximum variation sampling will be used to ensure recruitment of a diverse group of stakeholders (e.g., interdisciplinary, experience level, role in CwPAMS HPs) to explore their views and experiences regarding the effectiveness and implementation of the HSM. Stakeholder mapping will be conducted for each HP, and suitable participants will be invited to participate in either an interview or a focus group discussion.

### 2.4. Data Collection in Different Project Phases

The data for this study will be collected continuously throughout the project to support evaluation in three tiers: programme-level, partnership-level, and site-level. Data will be collected as a longitudinal time series and will be evaluated at three time points: pre-, mid- and post-implementation.

Pre-implementation phase

Collect pre-implementation feasibility data, that is, readiness of organisations (HPs) to implement the HSM.Collect baseline implementation data from each HP for each intervention, using indicators to measure progress.

Mid-term implementation phase:Collect mid-term implementation data from each HP using log-frame indicators to track progress on intended progress and outcome measures.Conduct a mid-term qualitative implementation evaluation to identify barriers and facilitators regarding implementation of the HSM while delivering AMS activities and interventions and develop real-time strategies via co-participatory approaches to overcome them.

Post-implementation phase:Collect final implementation data from each HP for each intervention, using indicators to track progress on intended measures (outputs, outcomes).Compare pre- and post-implementation data to evaluate the interventions.Collect and analyse post-implementation data to understand barriers and enablers of HSM and identify strategies to improve the progress and sustainment of AMS interventions in HPs.

### 2.5. Outcome Measures

Progress indicators have been designed and data collection tools will facilitate progress tracking against these measures. Table 1 shows an overview of outcome measures designed with respect to mandatory AMS interventions.

### 2.6. Informed Consent and Withdrawal from the Study

An information letter, provided at the time of consent, will include contact details for the research team, allowing participants to communicate their consent or withdrawal preference, or to find out more details about the study. Consent will be verbally received/recorded prior to participation in interview(s) and/or focus group(s) on the day of the data collection activity. Participants will have the right to withdraw from the study at any time before, during, or up to one week after interview or focus groups. While data cannot be retracted after this period, all efforts will be made to maintain participants’ anonymity. In quantitative tools and forms, there is a mandatory consent box to permit collection and use the data for research purposes. Participants who completed the tool and form will be considered to have given consent.

### 2.7. Data Preparation, Management and Storage

Data will be collected according to CPA’s data management and privacy policies and securely stored on CPA’s encrypted and secure cloud storage service; data will be anonymized by assignment of individual study codes for de-identification purposes.

### 2.8. Data Analysis

Quantitative data will be analysed using descriptive statistics with Microsoft Excel. SPSS software (V 22.0) will be used where needed for more advanced inferential statistical analysis.

Qualitative data will be transcribed and thematically analysed [21] using computer-assisted qualitative data analysis software (CAQDAS NVivo and ATLAS.ti). A deductive directed qualitative content analysis (DQlCA) [22] will be used, drawing on our theoretical guiding framework. In addition, we will conduct inductive analysis to capture emerging themes not currently represented in the framework and/or theories. To enhance trustworthiness, at least two experienced qualitative researchers (AI and GG—mixed method implementation science researchers with PhDs) will cross-code a portion of the transcripts independently and meet regularly to discuss disparities and agree on categorisation of data into the coding manual. Regular team meetings will be held to review codes, identify and discuss emerging patterns and themes to minimise bias and enhance rigour in data analysis.

### 2.9. Ethical Approval

This study is part of an internal programme evaluation and quality improvement; thus, no ethical approval has been deemed necessary.

## 3. Discussion

This protocol is a novel evaluation framework developed for AMS programmes using the HSM. This study is part of wider programmatic CPA and GHP work in eight African countries. Lessons learned from previous CwPAMS programmes prompted the teams to develop a strategic evaluation framework to offer support, as well as standardization, in order to evaluate progress in CwPAMS 2.0. Due to the broad nature of this evaluation protocol in capturing data from six different countries, this evaluation approach, along with tools and strategies to design AMS programmes, can be translated by other researchers to suit their particular needs. This would be particularly interesting to people currently facing AMR issues.

AMR remains a significant global health concern and is one of the leading causes of death worldwide, particularly in LMICs [13]. The challenge of AMR in LMICs is exacerbated by several interconnected factors. The misuse and overuse of antibiotics in healthcare, agriculture, animal care, and communities contributes significantly to the development of AMR [23]. Limited access to healthcare and diagnostics in LMICs often leads to empirical antibiotic treatment without accurate diagnosis, exacerbating the issue [13,23]. Inadequate infection prevention and control practices, along with poor sanitation and hygiene, further facilitate the spread of resistant bacteria. Moreover, restricted access to effective antibiotics may result in the use of less effective drugs or incorrect dosages, accelerating AMR emergence. Weak healthcare systems with insufficient surveillance, regulatory frameworks, and resources also hinder effective monitoring and containment efforts [24].

AMS activities are critical strategies aimed at improving the appropriate use of antimicrobial agents to reduce the risk of AMR globally, particularly in LMICs. These activities include developing evidence-based guidelines for antimicrobial use, educating healthcare providers and patients, monitoring prescribing patterns, implementing clinical protocols, conducting resistance surveillance, auditing practices, and promoting infection prevention. By optimising antimicrobial use and reducing unnecessary prescriptions, AMS initiatives aim to preserve antibiotic effectiveness, improve patient outcomes, and mitigate the growing threat of AMR [25]. However, achieving effective AMS in LMICs faces challenges, such as limited resources, inadequate knowledge and training, lack of up-to-date accessible guidelines, and difficulties in collecting actionable clinical data. The high burden of infectious diseases in these regions frequently leads to extensive antibiotic use, sometimes without prescription, contributing to the emergence and spread of antibiotic-resistant bacterial strains. Addressing AMR in LMICs requires comprehensive strategies, including strengthened healthcare infrastructure, enhanced infection control measures, and collaboration to ensure access to effective antibiotics while safeguarding their efficacy for future generations [14,25,26].

The World Health Organization (WHO) plays a crucial role in establishing global standards and formulating guidelines for the appropriate use of antimicrobials, advocating interdisciplinary care and One Health approaches [27]. However, it is equally important for LMICs to develop local solutions while implementing global and national recommendations to address AMR [14,27]. The CwPAMS initiative focuses on improving AMR surveillance, producing data on antimicrobial use, promoting responsible antimicrobial use, and enhancing the capacity of healthcare systems in Africa to combat AMR through shared learning among HPs. Through collaboration with UK higher education institutes, hospitals, and National Health Service (NHS) trusts, the CwPAMS programme provides technical support and customised strategies to address the specific challenges and contexts of AMS in African nations. By empowering local stakeholders with tailored strategies and technical support, CwPAMS aims to reduce the risks of AMR in their respective contexts [14].

To achieve this and maximise the resources available, an HSM was adopted to optimise resource utilisation and establish hubs as centers of excellence [15]. These centres of excellence are designed to effectively guide and initiate AMS interventions and actions across each region. Studies [28,29] inform us that implementing HSM in a healthcare context requires careful planning and coordination. Therefore, developing an implementation and evaluation protocol is crucial for providing a clear roadmap that ensures consistency, methodological rigour, and transparency throughout the project. Numerous implementation studies have developed study protocols that have helped in efficient resource and timeline management, establishing clear milestones and allocations to enhance project efficiency and minimise delays [30,31]. These protocols also ensure consistency and methodological rigour, enhancing credibility, transparency, and efficiency throughout the project. This proactive management approach minimises potential delays and enhances the overall fidelity of the programme. Implementation and evaluation protocols will also enhance scholarly transparency. Protocols may also facilitate the replication and verification of findings, promoting trust and confidence in project outcomes [31]. Publishing protocols as benchmarks allows for comparisons between planned methods and actual implementation, thereby bolstering research credibility and ensuring reproducibility while guarding against selective reporting bias in outcomes.

Numerous articles [15,32,33] have reported the application of HSM for improvement in disease and medicine management in healthcare settings. When well implemented, HSM can greatly enhance the efficiency and quality of healthcare delivery, making it a popular choice for large healthcare systems looking to optimise their services [32,33]. However, the application and efficacy of the HSM in the context of AMS have not yet been established. Under CwPAMS 2.0, we undertook a scoping review [34], to further understand the application of the HSM in the context of AMS and the tackling of AMR; the review identified only three studies in which this model has been applied, highlighting the great need for more research in this area. Although the data corpus was very small for inferring its utility in the context of AMS, the results were promising and have the capacity to influence researchers dealing with AMS in focusing on HSM to increase the implementation effectiveness of their interventions. The studies also identified several common barriers to implementing the HSM, including time constraints, financial concerns among participating bodies, technical difficulties, and limited support from upper management [15,16]. These challenges have been frequently reported in similar studies applying the HSM in various contexts and diseases [32,35,36]. Additionally, implementing the HSM faces hurdles such as managing cultural resistance to centralisation, ensuring effective communication across diverse units, and preventing potential congestion or over-reliance on the hub.

The scoping review guided us in developing a study protocol for our CwPAMS 2.0 implementation strategy [34]. This protocol aims to continuously monitor, evaluate, and promptly address implementation challenges in real time. It also facilitates the development of customised strategies that consider anticipated contextual obstacles. This protocol utilised a hybrid type II study design through shared HP insights and previous lessons learned from the CwPAMS programme, providing invaluable information for enhancing the adoption and sustainability of evidence-based practices and interventions [37]. By combining implementation strategies with intervention design and rollout, researchers can gain a comprehensive understanding of how interventions interact not only with each other but also with their implementation contexts [5]. This holistic approach facilitates a thorough assessment of factors influencing adoption rates and long-term sustainability.

One of the unique strengths of developing an evaluation framework as a research protocol lies in its bottom-up, HP-driven approach. This method emphasises co-participation and network reliance, allowing interventions to be customised by individuals intimately familiar with the system. While the log-frame and implementation indicators provide initial guidance and nudge the HPs in a specific direction, each HP has the flexibility to design and select interventions that address specific needs and align with their unique system requirements. Another strength of this study is that it involves diverse HPs, ranging from large hospitals to small health centres, academic universities, teaching hospitals, community pharmacies, and community health centres across six African countries. Notably, the hubs and spokes extend across both urban and suburban settings, allowing for the assessment of variations in patient populations and quality indicators, which will enable us to study the implementation of AMS interventions across these diverse contexts. Additionally, the mixed-methods approach, particularly qualitative interviews and case studies conducted throughout the study period, will yield valuable insights into the acceptability of and barriers to the HSM rollout. The study findings are expected to be relevant to similar programmes in other LMICs considering the adoption of this model for health interventions.

Our evaluation framework has some potential limitations. While our mixed-methods approach provides comprehensive data, the reliance on self-reported data from health partnerships may introduce reporting bias; nonetheless we mitigate this through triangulation with observational visits and multiple data sources. Data completeness may also vary across partnerships due to differences in resource availability, technical capacity, and competing priorities, which could affect the robustness of comparative analyses. Moreover, the drawbacks of developing protocols are that they can be time-consuming and require extensive planning and adaptation. This is relevant in projects that are time- or resource-constrained. This is due to the fact that real-world systems are complex and present unexpected challenges, where time often reveals unforeseen and unanticipated issues and challenges [38]. Although protocols are recommended, strict adherence may not always suit project implementation because real systems operate within dynamic and shifting contexts [5]. Implementing interventions in such complex ecosystems necessitates an approach informed by complexity theory, involving constant monitoring, evaluation, and readiness to modify implementation strategy as the project evolves [38]. We have designed the study protocol as a standardised guide and we have taken an adaptive approach that will allow the team to respond to changing circumstances. This might increase the likelihood of successful implementation and sustainability of the interventions. Due to the sheer number and complexity of the AMS interventions and variable implementation sites and contexts, our capacity to definitively attribute causality of outcomes to any individual intervention will likely be constrained, despite our comprehensive recording of progress measures across all domains. However, this project is broader in scope and does not specifically address interventional efficacy through causal mechanisms.

## 4. Conclusions

The evaluation framework is expected to guide, collect, and help evaluate the current understanding of the effectiveness and feasibility of implementing the HSM in AMS contexts. By aiming to optimise resource allocation and enhance healthcare efficiency, particularly in LMICs, we anticipate that the HSM, its supporting frameworks, and the implementation strategies developed will mark a significant and innovative advancement in addressing AMR. These efforts are anticipated to bridge gaps in AMS practices, not only within specific regions, but also in offering valuable insights applicable to diverse healthcare settings globally.

## Figures and Tables

**Figure 1 antibiotics-14-01218-f001:**
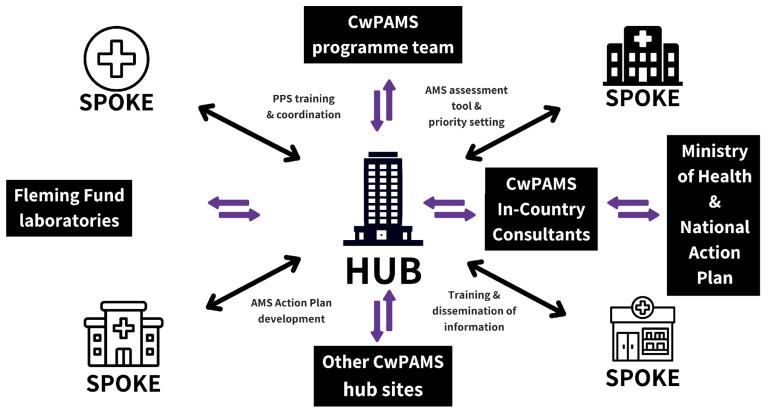
Diagrammatic presentation of the CwPAMS hub-and-spoke model.

**Figure 2 antibiotics-14-01218-f002:**
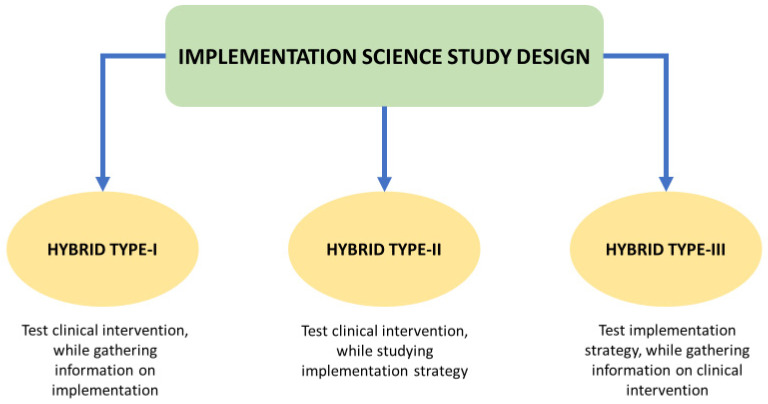
Implementation evaluation common study designs [17].

**Table 1 antibiotics-14-01218-t001:** Overview of CwPAMS 2.0 progress and outcome measures.

Evaluation Framework	Domains (Based on RE-AIM Framework)	Outcomes (OC)/Output (OP) *	Measures *	Data Collection Sources	Study Phases (Pre, Mid, Post)
Implementation evaluation framework	Reach	Uptake of interventions	(1) Participants delivering the interventions (2) Frequency of the intervention implemented (3) Interventions received, delivered by target audience	Quantitative and qualitative tools such as narrative reports, MEL portal, case studies, feedback surveys	Mid, post
	Effectiveness	Point Prevalence Survey (PPS) data, Practice change outcomes, behaviour change outcomes, PROMs	(1) Reduced antibiotic prescribing (2) Guidelines directed prescribing (3) Reduced overuse and misuse of antibiotic medicines (PPS data) (4) Development of antibiograms (5) Development of AMS guidelines, and other resources (6) Development of AMS committees (7) PPS conducted (8) PPS data used to inform practice (9) Development of AMS action plans (10) Use of reliable clinical information data to inform practices (11) Integration of interventions in HPs (12) Number of people trained (healthcare staff, public, communities) (13) Evidence of behaviour change	Quant and Qual tools such as pre- and post-assessments, narrative reports, MEL portal, case studies	Pre, mid, post
	Adoption	Utilisation	(1) Number of target audience receiving the intervention in the right dose (fidelity) (2) Number of target audience delivering the intervention in the right dose (fidelity) (3) Number of interventions integrated in HPs	Quant and Qual tools, narrative reports, MEL portal, case studies	Pre, mid and post
	Implementation	Feasibility Fidelity	(1) Processes and determinants of feasibility (2) Acceptability from stakeholders’ perspectives (both hub-and-spoke) (3) Number of target audience delivering the intervention in the right dose (fidelity- both hub-and-spoke) (4) Number of interventions targeted vs. actually delivered	Quant and qual tools, narrative reports, MEL portal, case studies	Pre, mid and post
	Maintenance	Sustainability	(1) Interventions still operational after funding period is over (3 months, 6 months) (2) Action undertaken/completed on AMS action plans (3) AMS committees still functional (regularly meet, record minutes, discusses data driven strategies and take actions for AMS) (4) HSM still being supported by organisations	Quant and qual post implementation tools, case studies	Post
Interventional/programmatic evaluation framework	Mandatory indicators				
	Infrastructure	OC1.1—Number of AMS action plans in place and implemented (excluding community pharmacies)		Quant and qual tools, narrative reports, MEL portal, case studies	Pre, post
	Data	OC1.2—Quality antimicrobial usage data is produced, analysed, shared, and used to develop relevant AMS interventions		Qual tools, Narrative reports, case studies	Pre, mid and post
	Data	OC1.3—Processes to integrate use of laboratory data (from Fleming Fund-funded labs where possible) into local AMS programmes and clinical practice are developed and strengthened		Qual tools, Narrative reports, case studies	Pre, mid and post
	Infrastructure	OC2.1—LMIC level: Evidence of how HPs are contributing towards national priorities and implementing the National Action Plans (NAPs) for AMR		Qual tools, Narrative reports, case studies	Post
	Infrastructure	OC2.2—LMIC level: Evidence of how national stakeholders will continue to support and sustain projects’ outcome (National Oversight Mechanisms implemented)		Qual tools, Narrative reports, case studies	Post
	Infrastructure	OC2.3—Regional level: Number of # LMIC hubs with established structures, leadership and AMS action plans in place to support activities and interventions amongst ‘spokes’/institutions		Quant and Qual tools, Narrative reports, MEL portal, case studies	Pre, mid and post
	Infrastructure	OC2.4—Institution level: Number of (%) LMIC healthcare institutions with fully operational AMS committees		Quant and Qual tools, Narrative reports, MEL portal, case studies	Pre, mid and post
	Infrastructure	OC2.5—Workforce level and Institutional level: Number of LMIC institutions with AMS/Infection Prevention and Control (IPC) guidelines, tools OR protocols embedded in their facility and being used effectively by health workers		Qual tools, Narrative reports, case studies	Pre, mid and post
	Volunteering	OC3.1—Number of UK volunteers who can name 5 barriers to functional AMS in LMICs as a result of participation in CwPAMS		Quant and Qual tools, Narrative reports, MEL portal, case studies	Mid and post
	Volunteering	OC3.2—Number of UK NHS staff volunteering days as a contribution to strengthen AMS in LMIC institutions		Quant and Qual tools, Narrative reports, MEL portal, case studies	Mid and post
	Volunteering	OC3.3—Number of UK institutions actively utilising volunteers’ skills and experiences in their own facility		Quant and Qual tools, Narrative reports, MEL portal, case studies	Post
	Training	OP 1.1—Number of LMIC healthcare staff trained by cadre and gender		Quant and Qual tools, Narrative reports, MEL portal, case studies	Pre, mid and post
	Behaviour	OP1.2—Number of (%) LMIC healthcare staff who have increased their capability, opportunity and motivation to undertake appropriate stewardship behaviours after attending CwPAMS training, by cadre and gender		Quant and Qual tools, Narrative reports, MEL portal, case studies	Pre, mid and post
	Data—PPS	OP1.4—Number of (%) LMIC institutions that have used PPS data to identify if interventions to improve antimicrobial prescribing practices are needed		Quant and Qual tools, Narrative reports, MEL portal, case studies	Pre, mid and post
	Infrastructure	OP 1.5—Number of LMIC institutions with new/updated AMS/IPC guidelines, tools or protocols in line with international or national guidelines/frameworks		Quant and Qual tools, Narrative reports, MEL portal, case studies	Pre, mid and post
	Training	OP 1.6—Number of HPs providing training on use of clinical microbiology data		Quant and Qual tools, Narrative reports, MEL portal, case studies	Pre, mid and post
	Training	OP 1.7—Number of (%) UK NHS staff surveyed who report an increase in knowledge and understanding of AMS in LMIC as a result of volunteering		Quant and Qual tools, Narrative reports, MEL portal, case studies	Mid and post
	Fellowships	OP 1.8—Number of Active fellowships		Quant and Qual tools, Narrative reports, MEL portal, case studies	Mid
	Fellowships	OP 1.9—Number of (%) Fellows making progress in professional/technical capabilities based on a combination of fellow’s (self-assessment) and mentors’ assessments		Quant and Qual tools, Narrative reports, MEL portal, case studies	Post
	GESI	OP1.10—Number of Projects with a Gender Equality and Social Inclusion (GESI) objective (Quant) and making progress e through a GESI-specific approach (Qual)		Quant and Qual tools, Narrative reports, MEL portal, case studies	Pre, mid and post
	Data	OP2.1—Number of HPs (Qual) (and institutions—Quant) where laboratory data is shared with (AMS Committees) clinical teams/IPC teams, and/or AMS, or guideline development groups.		Quant and Qual tools, Narrative reports, MEL portal, case studies	Pre, mid and post
	Data	OP2.2—Number of HPs that have developed a mechanism to share data between the facility and national stakeholders		Quant and Qual tools, Narrative reports, MEL portal, case studies	Mid and post
	Infrastructure	OP2.3—Evidence of institutional support for the uptake of locally derived AMS action plan		Qual tools, Narrative reports, case studies	Pre, mid and post
	Data	OP3.1—Number of antibiograms produced which are informed by local and up to date surveillance/microbiology data, by country		Quant and Qual tools, Narrative reports, MEL portal, case studies	Pre, mid and post
	Substandard and Falsified Medicines (SFMs)	OP3.2—Number of LMIC institutions reporting data specific to SFMs (i.e., SF antimicrobials)	Increased reporting of SFMs through national mechanisms	Quant and Qual tools, Narrative reports, MEL portal, case studies	Pre, mid and post
	Training	OP3.3—Number and type of information sharing and learning opportunities provided or facilitated by a) GHP/CPA to HPs; and b) between countries’ Hub and Spokes (by country)		Quant and Qual tools, Narrative reports, MEL portal, case studies	Pre, mid and post
	Data—PPS	OP3.4—Number (%) of LMIC institutions that have carried out a PPS (this includes collected and analysed PPS data) (Excluding some Outpatient Sites)		Quant and Qual tools, Narrative reports, MEL portal, case studies	Pre, mid
	Data—PPS	OP3.5—Number of PPS carried out across all institutions		Quant and Qual tools, Narrative reports, MEL portal, case studies	Pre, mid and post
	Data	OP3.6—Number of Peer reviewed publications of national level research (and media items coming from research)		Quant and Qual tools, Narrative reports, MEL portal, case studies	Post
	Infrastructure	OP3.7—Number of LMIC institutions that have updated or completed a (pre and post) AMS Assessment Tool		Quant and Qual tools, Narrative reports, MEL portal, case studies	Pre, post
	Training	OP4.1—Number of Awareness raising interventions amongst One Health groups and community pharmacies, by country and target group (patients, public, veterinary practitioners, community pharmacies etc.)		Quant and Qual tools, Narrative reports, MEL portal, case studies	Mid and post
	SFMs	OP4.2—Number of (%) LMIC teams with increased awareness of detection and/or reporting mechanisms for SFMs (antimicrobials)	Increased reporting of SFMs through national mechanisms	Quant and Qual tools, Narrative reports, MEL portal, case studies	Pre, post
	Infrastructure	OP4.3—Number of LMIC institutions that have, or included, a lab scientist/s (someone from the lab or microbiologist) in their AMS Team		Quant and Qual tools, Narrative reports, MEL portal, case studies	Pre, post
	Non-Mandatory indicators				
	Training	Number of LMIC healthcare staff trained in AMR/AMS		Quant and Qual tools, Narrative reports, MEL portal, case studies	Pre, mid and post
	Training	Number of LMIC healthcare staff trained in IPC		Quant and Qual tools, Narrative reports, MEL portal, case studies	Pre, mid and post
	Training	Number of LMIC healthcare staff trained in microbiology		Quant and Qual tools, Narrative reports, MEL portal, case studies	Pre, mid and post
	Training	Number of LMIC healthcare staff trained in good sample collection techniques		Quant and Qual tools, Narrative reports, MEL portal, case studies	Pre, mid and post
	Training	Number of LMIC healthcare staff trained in GESI		Quant and Qual tools, Narrative reports, MEL portal, case studies	Pre, mid and post
	Infrastructure	Number of Alcohol gel manufacturing facilities established		Quant and Qual tools, Narrative reports, MEL portal, case studies	Pre, post
	Infrastructure	Number of Active AMS Champions (train the trainer)		Quant and Qual tools, Narrative reports, MEL portal, case studies	Pre, post
	Data	Number of Antibiotic audits completed		Quant and Qual tools, Narrative reports, MEL portal, case studies	Pre, mid and post
	Data	Number of antibiotic audits interpreted/discussed and appropriate action(s) recommended		Quant and Qual tools, Narrative reports, MEL portal, case studies	Pre, mid and post
	Data	Number of (%) antibiograms requested prior to starting antibiotics		Quant and Qual tools, Narrative reports, MEL portal, case studies	Pre, mid and post
	Data	Number of antibiograms interpreted/discussed and appropriate action(s) recommended		Quant and Qual tools, Narrative reports, MEL portal, case studies	Pre, mid and post
	Volunteering	Number of volunteering days contributed by long-term volunteers/global health fellows		Quant and Qual tools, Narrative reports, MEL portal, case studies	Post

* MEL = Monitoring, Evaluation and Learning, GESI = Gender Equality and Social Inclusion, AMS = Antimicrobial Stewardship, LMIC = Low/Middle income country, AMR = Antimicrobial resistance, SFMs = Substandard and Falsified Medicines, PPS = Point Prevalence Survey, GHP = Global Health Partnership, CPA = Commonwealth Pharmacists’ Association, Quant = Quantitative, Qual = Qualitative, IPC = Infection, Prevention and control, NHS = National Health Services (UK), HSM = Hub-and-spoke Model, DQICA = Directed Qualitative Content Analysis, PROMs = Patient-reported outcome measures.

## Data Availability

Appendix A has been provided for additional information relevant to the manuscript. Further data/tools might be available for dissemination. Please contact the corresponding author to check the availability of data and materials relevant to this study.

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
