# Peer review of "A Hybrid Type II Hub-and-Spoke Model Evaluation Framework in the Commonwealth Partnerships for Antimicrobial Stewardship Programme—A Study Protocol"

_antibiotics, 2025, doi:10.3390/antibiotics14121218_

Round 1

Reviewer 1 Report

Comments and Suggestions for Authors

The manuscript provides significant knowledge through the presentation of a Hybrid Type II hub-and-spoke model evaluation framework within the Commonwealth Partnerships for Antimicrobial Stewardship (CwPAMS) programme. However, several issues should be addressed to improve the quality and clarity of the manuscript.

The study design and concept are well structured, but I have some concerns that require clarification.

In Figure 1, the term “CwPAMS” is shown with a red underline, indicating a detected spelling or formatting error in Microsoft Word. This should be corrected. Additionally, the figure legend “The hub and spoke model diagrammatic presentati.on” should be rechecked for grammatical accuracy. Consider revising to “Diagrammatic presentation of the hub and spoke model.”

In the Methodology section (2.2 Data collection), please clarify how irrelevant data were excluded. Provide explicit criteria or a description of the process for excluding non-related data.

Table 1, titled “Overview of CwPAMS 2.0 progress and outcomes measures,” presents a large volume of data that is difficult to interpret. It is suggested to divide the table into 2–3 smaller tables or use graphical representations such as flow charts or pie charts to enhance readability and comprehension.

Include a limitations paragraph at the end of the Discussion section to acknowledge potential constraints of the study, such as data completeness, representativeness, or scope.

Review the References section carefully. The current reference formatting does not follow the journal’s guidelines. Ensure consistency in style, punctuation, and citation order according to the journal’s requirements.

Overall, addressing these issues will improve the manuscript’s presentation and readability.

Author Response

The manuscript provides significant knowledge through the presentation of a Hybrid Type II hub-and-spoke model evaluation framework within the Commonwealth Partnerships for Antimicrobial Stewardship (CwPAMS) programme. However, several issues should be addressed to improve the quality and clarity of the manuscript.

The study design and concept are well structured, but I have some concerns that require clarification. 

In Figure 1, the term “CwPAMS” is shown with a red underline, indicating a detected spelling or formatting error in Microsoft Word. This should be corrected. Additionally, the figure legend “The hub and spoke model diagrammatic presentati.on” should be rechecked for grammatical accuracy. Consider revising to “Diagrammatic presentation of the hub and spoke model.”

Response: We thank the reviewer for pointing out this formatting issue. The red underline was a Microsoft Word spell-check artifact and has been removed. We have updated Figure 1 to eliminate all spelling/formatting indicators.

The Figure 1 legend has been revised from "The CwPAMS hub and spoke model diagrammatic presentation" to "Diagrammatic presentation of the CwPAMS hub and spoke model" as suggested, providing a clearer and more grammatically correct caption.

In the Methodology section (2.2 Data collection), please clarify how irrelevant data were excluded. Provide explicit criteria or a description of the process for excluding non-related data.

Response: Thank you for providing this suggestion. We have enhanced Section 2.2. The following has been added” All data collected will be screened against predefined inclusion criteria aligned with our logframe indicators (Table 1) and RE-AIM framework domains. Data will be considered irrelevant and excluded if it: (1) does not align with any of the mandatory or non-mandatory outcome measures defined in our evaluation framework, or (2) lacks sufficient detail to contribute meaningfully to our implementation evaluation. The research team (AI and GG) will review all collected data during monthly meetings to ensure alignment with study objectives, and any ambiguous cases or unusual data will be discussed with the broader team for consensus decision-making”

Table 1, titled “Overview of CwPAMS 2.0 progress and outcomes measures,” presents a large volume of data that is difficult to interpret. It is suggested to divide the table into 2–3 smaller tables or use graphical representations such as flow charts or pie charts to enhance readability and comprehension.

Response: We appreciate the reviewer's concern about Table 1's complexity. We acknowledge that the comprehensive nature of our evaluation framework necessitates presenting substantial information. We have now carefully revised Table 1 to enhance readability while presenting the framework.

Include a limitations paragraph at the end of the Discussion section to acknowledge potential constraints of the study, such as data completeness, representativeness, or scope.

Response: Thank you for this suggestion. We have now modified the limitation section and have added the following” Our evaluation framework does have potential limitations. While our mixed-methods approach provides comprehensive data, the reliance on self-reported data from health partnerships may introduce reporting bias; nonetheless we mitigate this through triangulation with observational visits and multiple data sources. Data completeness may also vary across partnerships due to differences in resource availability, technical capacity, and competing priorities, which could affect the robustness of comparative analyses.

Review the References section carefully. The current reference formatting does not follow the journal’s guidelines. Ensure consistency in style, punctuation, and citation order according to the journal’s requirements.

Response: We sincerely apologize for the inconsistencies in our reference formatting. We have thoroughly reviewed and revised all references to comply with Antibiotics journal guidelines.

Reviewer 2 Report

Comments and Suggestions for Authors

Dear authors,

congratulations for your valuable work. Please, find here below some suggestions in order to further improve the quality of your paper.

  1. Please, avoid acronyms in the title.
  2. I think a deeper analysis of already existing models in AMS policies around the world could improve the context presentation in the introduction. As examples, here are some interesting references on the topic: https://pubmed.ncbi.nlm.nih.gov/40005771/; https://pubmed.ncbi.nlm.nih.gov/35739564/; https://pubmed.ncbi.nlm.nih.gov/38179856/
  3. Please, provide a rigorous presentation of the surveys you mentioned in the 2.4 subchapter.
  4. Since you reported your study period is limited from March 2023 to March 2025, it means you have conducted a retrospective study. Am I right? If I understand well, I will recommend to revise the study design section reporting that you are currently presenting retrospective data.

Author Response

Dear authors,

congratulations for your valuable work. Please, find here below some suggestions in order to further improve the quality of your paper.

Response: Thankyou for your kind words.

  1. Please, avoid acronyms in the title.

Response: We appreciate this suggestion for improving accessibility to a broader readership.We have revised the manuscript title:

"A Hybrid Type II Hub and Spoke Model Evaluation Framework in the Commonwealth Partnerships for Antimicrobial Stewardship Programme”

  1. I think a deeper analysis of already existing models in AMS policies around the world could improve the context presentation in the introduction. As examples, here are some interesting references on the topic: https://pubmed.ncbi.nlm.nih.gov/40005771/; https://pubmed.ncbi.nlm.nih.gov/35739564/; https://pubmed.ncbi.nlm.nih.gov/38179856/

Response: We thank the reviewer for these valuable references and the suggestion to strengthen our contextual background. We appreciate the notion that situating our work within the broader landscape of global AMS to further contextualize our work with recent analyses of AMS governance frameworks and organizational models as they play a critical role in shaping stewardship policies worldwide. However, our study purposefully takes a different approach by evaluating a hub and spoke model to deliver AMS interventions. This represents more of a tool rather than policies. The primary aim of this model is to empower health partnerships with flexibility in designing and implementing interventions.

In our programme, each participating country and partnership was encouraged to align interventions with mandatory indicators while adapting action plans to their own national strategies and local realities. Many of these MEL plans draw inspiration from WHO recommendations. This decentralized design explicitly allowed for interventions that are responsive to specific institutional environments, varied operational challenges, and unique antimicrobial resistance priorities. However, a systematic comparison with governance-based models promises to yield valuable insights, particularly regarding structural and policy-level influences, such an analysis is beyond the scope of the current protocol. We intend to add these references in our forthcoming results manuscript, where we will examine the impact of governance structures on stewardship outcomes.

We appreciate the reviewer’s constructive input and we believe this will strengthen our future work.

  1. Please, provide a rigorous presentation of the surveys you mentioned in the 2.4 subchapter.

Response: We have now included quantitative surveys/data collection tool templates in Appendix 6 (supplementary file). We have also cited this in the main manuscript under section 2.4.

  1. Since you reported your study period is limited from March 2023 to March 2025, it means you have conducted a retrospective study. Am I right? If I understand well, I will recommend to revise the study design section reporting that you are currently presenting retrospective data.

Response: We thank the reviewer for this important clarification request. As this is not a retrospective study, this manuscript presents a study protocol for a prospective, longitudinal hybrid implementation evaluation. The study began in March 2023 and concluded in March 2025. As of the manuscript submission date we are in the midst of data analysis and the findings will be published in due time. We have provided more clarity now under section 2.2.

Reviewer 3 Report

Comments and Suggestions for Authors

The authors describe an interesting model for evaluating antimicrobial stewardship but do not show the results of the programme's implementation, even with preliminary data. It would be interesting to know how effective the model is.

Author Response

The authors describe an interesting model for evaluating antimicrobial stewardship but do not show the results of the programme's implementation, even with preliminary data. It would be interesting to know how effective the model is.

Response: We sincerely thank the reviewer for this valuable feedback and for recognizing the interest in our evaluation model. We appreciate the desire to see implementation results, however this manuscript is a study protocol paper, not a results paper. We are currently compiling the results paper and the findings will soon be submitted to the same journal, which provides an opportunity to read the results of this model for interested readers.